# The impact of a routine late third trimester growth scan on the incidence, diagnosis, and management of breech presentation in Oxfordshire, UK: A cohort study

**Ibtisam Salim**[1,2]*, **Eleonora Staines-Urias**[1], **Sam Mathewlynn**[2], **Lior Drukker**[1], **Manu Vatish**[1], **Lawrence Impey**[2]

**1** Nuffield Department of Women's Reproductive Health, John Radcliffe Hospital, Oxford University, Oxford, United Kingdom, **2** Oxford Fetal Medicine Unit, John Radcliffe Hospital, Headley Way, Oxford, United Kingdom

* Ibtisam.salim@wrh.ox.ac.uk

**Data Availability Statement:** All relevant data are within the manuscript and its Supporting Information files.

## Abstract

### Background

Breech presentation at term contributes significantly to cesarean section (CS) rates worldwide. External cephalic version (ECV) is a safe procedure that reduces term breech presentation and associated CS. A principal barrier to ECV is failure to diagnose breech presentation. Failure to diagnose breech presentation also leads to emergency CS or unplanned vaginal breech birth. Recent evidence suggests that undiagnosed breech might be eliminated using a third trimester scan. Our aim was to evaluate the impact of introducing a routine 36-week scan on the incidence of breech presentation and of undiagnosed breech presentation.

### Methods and findings

We carried out a population-based cohort study of pregnant women in a single unit covering Oxfordshire, United Kingdom. All women delivering between 37+0 and 42+6 weeks gestational age, with a singleton, nonanomalous fetus over a 4-year period (01 October 2014 to 30 September 2018) were included. The mean maternal age was 31 years, mean BMI 26, 44% were nulliparous, and 21% were of non-white ethnicity. Comparisons between the 2 years before and after introduction of routine 36-week scan were made for 2 primary outcomes of (1) the incidence of breech presentation and (2) undiagnosed breech presentation. Secondary outcomes related to ECV, mode of birth, and perinatal outcomes. Relative risks (RRs) with 95% confidence intervals (CIs) are reported. A total of 27,825 pregnancies were analysed (14,444 before and 13,381 after). A scan after 35+0 weeks was performed in 5,578 (38.6%) before, and 13,251 (99.0%) after ($p < 0.001$). The incidence of breech presentation at birth did not change significantly (2.6% and 2.7%) (RR 1.02; 95% CI 0.89, 1.18; $p = 0.76$). The rate of undiagnosed breech before labour reduced, from 22.3% to 4.7% (RR 0.21; 95% CI 0.12, 0.36; $p < 0.001$). Vaginal breech birth rates fell from 10.3% to 5.3% (RR 0.51; 95% CI 0.30, 0.87; $p = 0.01$); nonsignificant increases in elective CS rates and

**Funding:** The authors received no specific funding for this work.

**Competing interests:** The authors have declared that no competing interests exist.

**Abbreviations:** CI, confidence interval; CPR, cerebroplacental ratio; CS, cesarean section; ECV, external cephalic version; EDD, expected date of delivery; OUH, Oxford University Hospitals; OxGRIP, Oxford Growth Restriction Identification Programme; PI, pulsatility index; RR, relative risk; STROBE, Strengthening the Reporting of Observational Studies in Epidemiology.

decreases in emergency CS rates for breech babies were seen. Neonatal outcomes were not significantly altered. Study limitations include insufficient numbers to detect serious adverse outcomes, that we cannot exclude secular changes over time which may have influenced our results, and that these findings are most applicable where a comprehensive ECV service exists.

## Conclusions

In this study, a universal 36-week scan policy was associated with a reduction in the incidence but not elimination of undiagnosed term breech presentation. There was no reduction in the incidence of breech presentation at birth, despite a comprehensive ECV service.

## Author summary

### Why was this study done?

- The risk of complications at birth are higher in babies that are presenting breech, which is therefore a common reason for planning a cesarean section. In about a third, however, breech presentation is not discovered until labour. This leads to unanticipated cesarean birth or vaginal breech birth.

- Recent data suggest that ultrasound screening in nulliparous women could virtually eliminate undiagnosed breech presentation but did not assess the effect in multiparous women who comprise more than 50% of all births.

- Given that known breech presentation can often be corrected before birth, improved antenatal detection through screening could also lead to a lower incidence of term breech presentation, but this has never been tested.

### What did the researchers do and find?

- This study assessed breech outcomes of 278,250 term births before and after the introduction of a third trimester screening ultrasound, which increased the proportion of pregnancies screened from 39% to 99%.

- The incidence of undiagnosed breech presentation fell substantially but 1 in 20 breech babies at term were still not diagnosed until labour.

- The incidence of breech presentation at term did not reduce, albeit from an already low incidence.

### What do these findings mean?

- Introducing a third trimester screening scan into a maternity service is unlikely to prevent the occasional need for skills at breech birth, which themselves require experience.

- That the incidence of term breech presentation did not decline with screening limits the latter's usefulness, but where services for correction of breech presentation require and receive improvement, an effect might be anticipated.

- Improvements in neonatal outcomes should not be assumed because of reduced expertise in management of those that still present as undiagnosed breech and with women exercising maternal preference.

## Introduction

Breech presentation has an incidence at term of 3% to 4% and is more common among preterm births [1]. Vaginal breech birth has been associated with higher neonatal morbidity and mortality than cesarean (CS) [2] or cephalic birth. In an attempt to reduce birth-related complications, most breech babies are delivered by CS and thus breech babies make a substantial contribution to CS rates in developed countries [3]. Undergoing CS leads to both short- and long-term maternal complications [4] and may even be associated with long-term health consequences like obesity and diabetes for the newborn [4–6]. CS is costly and has financial implications for healthcare providers, which can be influenced by whether the procedure is elective or as an emergency [7].

If diagnosed antenatally, term breech presentation can often be corrected using external cephalic version (ECV) where the fetus is manipulated through the abdominal wall to a cephalic presentation. This is successful in nearly 50% of attempts; reversion to breech is rare and leads to a reduction in breech presentation at term [8]. The procedure is considered safe [8–10]. Those in whom ECV is declined, fails, or is contraindicated can be appropriately counselled regarding birth.

While antenatal assessment of presentation is a cornerstone of antenatal care, approximately one-third of breech babies are not diagnosed antenatally [11]. The sensitivity of clinical examination alone is reported as 70% [12]. This failure to diagnose breech presentation antenatally reduces the opportunity to correct it via ECV. It also means that undiagnosed breech babies may present in advanced labour and that CS is often performed as an emergency procedure. Further, it reduces the opportunity to appropriately counsel and prepare for those women desiring a vaginal birth.

Breech presentation is easily diagnosed using ultrasound. Given that spontaneous version from breech to cephalic occurs after 36 weeks in <10% [13], and reversion after ECV in less than 3% [10], universal ultrasound at late gestation should improve the rate of antenatal diagnosis and, logically, lead to a reduction in the incidence of breech presentation at term. In the United Kingdom, United States of America, and many European countries, third trimester ultrasound is not performed as routine clinical practice, although this trend is changing, with the primary clinical driver being attempting to detect babies at increased risk of stillbirth [14]. However, a corollary of this is an assessment of presentation by ultrasound at the time of fetal risk assessment.

We noted the findings of a large recent UK study that offered nulliparous women a universal scan at 36 weeks gestation [15]: The authors suggested that use of universal scanning at this gestation could virtually eliminate undiagnosed term breech presentation. Unfortunately, the findings cannot be applied to multiparous women who make up half the pregnant population but have a higher incidence of spontaneous version [13]. Because of this, and the relatively

poor usage and success rate (14%) of ECV in the study, a further question remains unanswered. This is whether universal ultrasound, by allowing more usage of ECV, leads to a reduction in the incidence of term breech presentation.

From September 2016, a universal 36-week scan was offered to all women giving birth in Oxford University Hospitals NHS Trust (OUH). The aim of our study was to determine the "real world" impact of this policy on breech presentation at term.

## Methods

This is a population-based cohort study of 4 years of births in a large UK teaching hospital, assessing the impact of a major change in unit policy: a universally offered late third trimester scan. Outcomes were prespecified (S1 Analysis plan). Primary outcomes were (1) the overall incidence of breech presentation and (2) the incidence of undiagnosed breech presentation. Secondary outcomes related to ECV clinic outcomes, mode of birth for breech presentation, and perinatal outcomes.

Pregnancies with an expected date of delivery (EDD) between 01 October 2014 and 30 September 2016 were compared those with an EDD of 01 October 2016 to 30 September 2018. All women delivering in OUH with a singleton, nonanomalous fetus were included, and the term group, women giving birth from 37+0 weeks' gestation, were included in analysis.

All data were collected prospectively, with output from respective data collection systems: electronic patient record (Cerner Millennium) for maternity data, Badgernet (Clevermed) for neonatal data, Viewpoint (GE Healthcare) for ultrasound data, and prospective breech clinic records were merged. These were analysed using Stata 15 software (StataCorp, College Station, Texas). The Strengthening the Reporting of Observational Studies in Epidemiology (STROBE) guidelines were followed (S1 STROBE Checklist). Demographic characteristics, pregnancy, birth, and neonatal outcomes were summarised in the 2 groups with mean and standard deviation for continuous variables and count and proportion for categorical variables and compared by means of $t$ test or chi-squared test as appropriate. Where missing values occurred, which was only among demographics, calculations were performed using only pregnancies with data as the denominator. Differences between the 2 groups were compared using relative risk (RR) ratio, calculated as the percentage of pregnancies with the outcome of interest after the introduction of the 36-week scan, divided by the percentage of pregnancies with the outcome of interest prior to its introduction, with 95% confidence intervals (CIs).

OUH is a large tertiary referral unit in the UK. A specialist-dedicated "breech clinic" has been in place since 1998. The normal care pathway is that all women from 35 weeks with an ultrasonographically proven breech presentation, or where breech presentation is suspected clinically, are referred unless a preexisting indication for CS exists. Nulliparous women are seen from 36+0 weeks; multiparous women from 37+0 weeks. Ultrasound is performed in the clinic, and women are assessed for their suitability for ECV according to published criteria [10] and which, if appropriate and accepted, is performed in the same clinic as described elsewhere [10]. The procedure itself or the operators did not alter over the period under study. In persistent breech presentations, the mode of birth is discussed: Women are assessed as described elsewhere [16] for their suitability for vaginal breech birth, which is planned depending on maternal choice. Alternatively, elective CS birth is offered.

In October 2016, it became unit policy that all women with a singleton fetus were offered a routine ultrasound scan between 35+0 and 36+6 weeks' gestation, as part of a service initiative (Oxford Growth Restriction Identification Programme, OxGRIP). Prior to this, ultrasound scans were arranged ad hoc according to perceived clinical risk. The 36-week scan was prebooked at the anomaly scan, or later if the anomaly scan had not taken place in OUH. At the

36-week scan, the presentation and placental site were determined, the deepest pool of amniotic fluid and fetal biometry were measured, and the umbilical artery and middle cerebral artery Doppler pulsatility index (PI) assessed to calculate the cerebroplacental ratio (CPR). Where breech presentation was noted, women were referred directly to the breech clinic for an appointment from 36 or 37 weeks' gestation according to their parity.

For the purposes of the analysis, different subgroups were created. A flow diagram (Fig 1) demonstrates the possible flows of breech babies in the third trimester to aid understanding of these cohorts.

1. "Breech presentation at birth" was defined as where the fetus delivered breech or, if delivery had been by CS, this was the presentation at the start of surgery.

2. "Undiagnosed breech" was defined as where the presentation at birth ($\geq$37+0 weeks) was breech, but the presentation had not been recorded at any ultrasound examination performed from 34+0 weeks and there had been no referral to the breech clinic, or there had been no elective CS with breech given as a primary or secondary indication or, if an ECV had been successful, breech presentation had not been recorded at any ultrasound following this.

3. "Breech in the third trimester" was a surrogate definition created to allow neonatal outcome data to take account of the risks of ECV as well as being breech presentation at birth. It was defined as breech presentation at birth, or at any scan from 35+0 weeks or where a successful ECV had been performed.

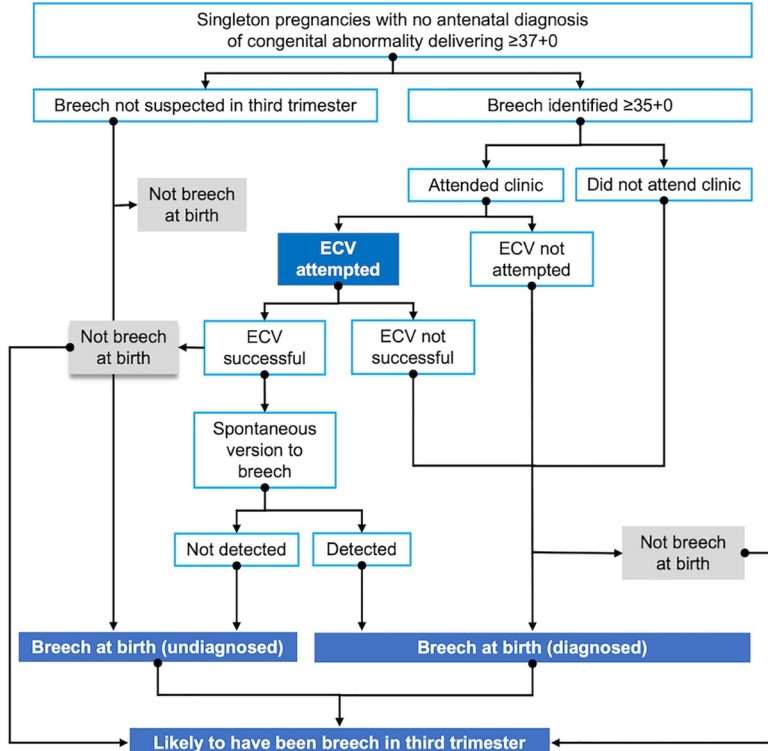

**Fig 1. Pathways of the breech fetus in the third trimester. ECV, external cephalic version.**

### Ethics statement

The ethics approval was granted by the Health Research Authority, IRAS project ID 222260 and REC reference: 17/SC/0374 on 27/07/2017. Informed consent was not required as the analysis was of routinely collected patient data. The study is reported as per the STROBE guideline (S1 STROBE Checklist).

## Results

The cohort comprised 29,559 births after 24 completed weeks. Of these, 27,825 (94.1%) were at 37+0 weeks or beyond, of which 14,444 were born prior to the universal scan policy and 13,381 after its introduction. There were 740 breech presentations at birth from 37+0 weeks, of whom 58 (7.8%) delivered vaginally. There were a further 217 preterm breeches, of whom 57 (26.3%) delivered vaginally. Demographic details are shown in Table 1: There were small but significant increases in mean BMI and maternal age. The number of women delivering at term undergoing any growth scan between 35+0 weeks and birth increased from 5,578 (38.6%) to 13,251 (99.0%) ($p < 0.001$).

Table 1 describes the demographic characteristics of the cohort and demonstrates the increased usage of ultrasound following the introduction of the universal scan.

The incidence of breech presentation at term birth did not reduce (2.6% versus 2.7%; RR 1.02, 95% CI 0.89, 1.18; $p = 0.76$) (Table 2). The incidence of undiagnosed breech presentation (Table 3) was reduced from significantly 25.5% to 8.3% (RR 0.33; 95% CI 0.22, 0.48; $p < 0.001$), and in the group who actually laboured, from 22.5% to 4.7% (RR 0.21; 95% CI 0.13, 0.35; $p < 0.001$).

The unaltered incidence of breech presentation occurred despite an increase in the number of breech babies seen in the clinic (Table 3) and a possible increase in the overall number of successful ECVs (RR 1.20; 95% CI 0.96, 1.51; $p = 0.10$). There were nonsignificant increases in eligible women declining ECV, from 12.1% to 13.7% (RR 1.14; 95% CI 0.78, 1.66), and

**Table 1. Demographic characteristics of the study population.**

|  | Pre-universal 36-week scan | Post-universal 36-week scan |
|---|---|---|
| N (total) | 15,371 | 14,188 |
| Deliveries ≥24 and <37 weeks, *n (%)* | 923 (6.0) | 802 (5.7) |
| Breech at delivery <37 weeks, *n (%)* | 107 (0.7) | 110 (0.8) |
| Deliveries ≥37 and <43 weeks, *n (%)* | 14,444 (94.0) | 13,381 (94.3) |
| Deliveries ≥43 weeks, *n (%)* | 4 (0.03) | 5 (0.04) |
| Maternal age (years), *mean ± SD* | 30.8 (5.5) | 31.1 (5.4) |
| Body mass index (kg/m$^2$), *mean ± SD* | 25.4 (5.5) | 25.7 (5.6) |
| Nulliparous, *n (%)* | 6,699 (43.6) | 6,271 (44.2) |
| Parity >3, *n (%)* | 386 (2.5) | 315 (2.2) |
| White ethnicity, *n (%)* | 12,143 (79.0) | 11,274 (79.5) |
| Gestational age at delivery (weeks), *mean ± SD* | 39.6 (2.0) | 39.7 (2.0) |
| Male fetus, *n (%)* | 7,912 (51.5) | 7,345 (51.8) |
| Birthweight (g), *mean ± SD* | 3,434.7 (586.7) | 3,446.3 (590.9) |
| Ultrasound scan ≥35+0 weeks among term deliveries, *n (%)* | 5578 (38.6) | 13251 (99.0) |

*$p$-Value from $t$ test or Pearson chi-squared test, as appropriate.

Missing data were 2% for body mass index and 7.8% for maternal ethnicity; <2% otherwise.

**Table 2. Incidence, diagnosis, and mode of birth of term breech babies.**

| | Pre-universal 36-week scan | Post-universal 36-week scan | RR (95% CI) | *p*-value |
|---|---|---|---|---|
| **N (total)** | **14,444** | **13,381** | | |
| Breech presentation at birth, *n (%)* | 380 (2.6) | 360 (2.7) | 1.02 (0.89–1.18) | 0.76 |
| Among all breech presentations at birth: | 380 | 360 | | |
| Planned vaginal breech birth, *n (%)* | 13 (3.4) | 22 (6.1) | 1.79 (0.91, 3.49) | 0.09 |
| Actual vaginal breech birth, *n (%)* | 39 (10.3) | 19 (5.3) | 0.51 (0.30, 0.87) | 0.01 |
| Elective CS, *n (%)* | 218 (57.4) | 241 (66.9) | 1.17 (1.04, 1.31) | 0.01 |
| Emergency CS, *n (%)* | 123 (32.4) | 100 (27.8) | 0.86 (0.69, 1.07) | 0.18 |
| Diagnosed before birth, *n (%)* | 283 (74.5) | 330 (91.7) | 1.23 (1.15, 1.32) | <0.001 |
| Gestational age at delivery (weeks), *mean ± SD* | 39.1 (0.9) | 39.0 (0.8) | | 0.02 |
| Planned vaginal breech birth, *n (%)* | 13 (4.6) | 22 (6.7) | 1.45 (0.74, 2.83) | 0.27 |
| Actual vaginal breech birth, *n (%)* | 14 (5.0) | 17 (5.2) | 1.04 (0.52, 2.07) | 0.91 |
| Elective CS, *n (%)* | 206 (72.8) | 228 (69.1) | 0.95 (0.86, 1.05) | 0.31 |
| Emergency CS, *n (%)* | 63 (22.3) | 85 (25.8) | 1.16 (0.87, 1.54) | 0.32 |
| Breech undiagnosed before birth, *n (%)*** | 97 (25.5) | 30 (8.3) | 0.33 (0.22, 0.48) | <0.001 |
| Gestational age at delivery (weeks), *mean ± SD* | 39.7 (1.3) | 39.5 (1.2) | | 0.44 |
| Planned vaginal breech birth, *n (%)* | 0 (0.0) | 0 (0.0) | NA | NA |
| Actual vaginal breech birth, *n (%)* | 25 (25.8) | 2 (6.7) | 0.26 (0.07, 1.03) | 0.06 |
| Elective CS, *n (%)* | 12 (12.4) | 13 (43.3) | 3.50 (1.79, 6.84) | <0.001 |
| Emergency CS, *n (%)* | 60 (61.9) | 15 (50.0) | 0.81 (0.55, 1.19) | 0.29 |
| Breech undiagnosed before labour***, *n(%)* | 85 (22.3) | 17 (4.7) | 0.21 (0.12, 0.36) | <0.001 |

*\*p*-Value from *t* test.

\*\*Defined as breech presentation at delivery *and* either (1) no scan ≥35+0 weeks showing breech; (2) successful external cephalic version; or (3) no elective CS with breech presentation as primary/secondary indication.

\*\*\*Labour started with undiagnosed breech: elective CS excluded.

CI, confidence interval; CS, cesarean section; NA, not applicable; RR, relative risk.

decreases in the success rate (49.5% to 45.3%) (RR 0.92; 95% CI 0.78, 1.08; *p* = 0.29) of ECV attempts, although not in the number of women where ECV was considered contraindicated (RR 0.69 95% CI 0.37, 1.30; *p* = 0.25).

**Table 3. ECV in babies delivering at term.**

| | Pre-universal 36-week scan | Post-universal 36-week scan | RR (95% CI) | *p*-value |
|---|---|---|---|---|
| **N (total)** | **14,444** | **13,381** | | |
| Attendances at ECV clinic, *n (%)* | 952 (6.6) | 650 (4.9) | 0.74 (0.67, 0.81) | <0.001 |
| Breech presentation at clinic visit, *n (% of attendances)* | 356 (37.4) | 436 (67.1) | 1.79 (1.63, 1.98) | <0.001 |
| Breech presentation at clinic, *n (% of term deliveries)* | 356 (2.5) | 436 (3.3) | 1.32 (1.15, 1.51) | <0.001 |
| ECV attempted if breech at clinic visit, *n (%)* | 299 (84.0) | 364 (83.5) | 0.99 (0.93, 1.06) | 0.85 |
| Successful, *n (%of attempts)* | 148 (49.5) | 165 (45.3) | 0.92 (0.78, 1.08) | 0.28 |
| Successful, *n (% of term deliveries)* | 148 (1.0) | 165 (1.2) | 1.20 (0.96, 1.51) | 0.10 |
| Unsuccessful, *n (%)* | 151 (50.5) | 199 (55.7) | 1.08 (0.94, 1.25) | 0.29 |
| ECV not attempted, *n (%)* | 57 (16.0) | 72 (16.5) | 1.03 (0.75, 1.42) | 0.85 |
| Contraindicated, *n (%)* | 16 (28.1) | 14 (19.4) | 0.69 (0. 37, 1.30) | 0.25 |
| Declined, *n (%)* | 41 (71.9) | 58 (80.6) | 1.12 (0.92, 1.37) | 0.26 |
| ECV not attempted OR unsuccessful, *n (%)* | 208 (58.4) | 271 (62.2) | 1.06 (0.95, 1.19) | 0.29 |

CI, confidence interval; ECV, external cephalic version; RR, relative risk.

**Table 4. Perinatal outcomes of pregnancies with a third trimester\* breech presentation.**

|  | Pre-universal 36-week scan | Post-universal 36-week scan | RR (95% CI) | p-value |
|---|---|---|---|---|
| N (total) | 527 | 525 |  |  |
| Extended perinatal mortality, n (%) | 2 (0.4) | 0 (0.0) | NA | NA |
| NCAO, n (%) | 44 (8.4) | 40 (7.6) | 0.91 (0.61, 1.38) | 0.67 |
| Hypoxic-ischaemic encephalopathy 1–3, n (%) | 6 (1.1) | 1 (0.2) | 0.17 (0.02, 1.38) | 0.10 |
| NNU admission, n (%) | 35 (6.6) | 31 (5.9) | 0.88 (0.56, 1.42) | 0.62 |
| Apgar <7 at 5 min, n (%) | 10 (2.0) | 6 (1.2) | 0.59 (0.22, 1.61) | 0.30 |
| Birthweight <3$^{rd}$ centile, n (%) | 3 (0.6) | 4 (0.8) | 1.34 (0.30, 5.95) | 0.70 |

\*Combines breech at delivery and pregnancies that underwent successful external cephalic version for breech presentation.

NCAO, neonatal composite adverse outcome, including Apgar at 5 minutes <7, cord arterial pH <7.1, admission to neonatal intensive care unit, perinatal death (stillbirth or death within 28 days of life), or hypoxic-ischaemic encephalopathy Grade 1–3; NA, not applicable; NNU, neonatal unit.

The proportion of vaginal breech births reduced from 10.3% to 5.3% (RR 0.51; 95% CI 0.30, 0.87; p = 0.01), largely due to the reduction in undiagnosed breech presentation in labour; this may also be responsible for the possible increase in planned breech birth. Although the proportion delivered by elective CS was significantly increased (57.4% to 66.9%, RR 1.17; 95% CI 1.04, 1.31; p = 0.01), the reduction in emergency CS, from 32.4% to 27.8% (RR 0.86; 95% CI 0.69, 1.07; p = 0.18), was smaller.

Of the 17 term babies (Table 2) who started labour unknown to be breech after the 36-week scan policy started, one (5.9%) had not had a scan, 13 (76.5%) were multiparous, and only 2 (11.8%) had been transverse or oblique at the 36-week scan. None of these babies had had a successful ECV where reversion to breech occurred, but this did occur in 1 pregnancy where it was subsequently diagnosed before birth. Table 4 examines morbidity and mortality in a manner that encompasses any risks of ECV and therefore using not just term breech presentation, but a surrogate for babies that were probably breech in the third trimester. There was a nonsignificant reduction in all markers of neonatal morbidity and the numbers of extended perinatal deaths (still birth or neonatal death <28 days of life) reduced from two to zero. One death was an intrapartum death with an unsupervised, unplanned birth at home of an undiagnosed breech baby; the other had been diagnosed but was an antepartum stillbirth.

## Discussion

We observed that the introduction of a universal third trimester ultrasound scan across an entire, large maternity unit was followed by a large reduction in the incidence of undiagnosed breech presentation but not its elimination: About 5% of breech presentations were still only detected in labour. Vaginal birth among breech babies was reduced: Before the universal scan policy, more than half of all actual vaginal breech births occurred in women with an undiagnosed breech presentation. CS birth for breech increased, although the reduction in the proportion performed as an emergency was relatively small. There was no alteration in the incidence of breech presentation at term despite a greater opportunity to use ECV to prevent it. Adverse neonatal outcomes were rare, and a nonsignificant benefit was observed.

In 2019, Wastlund and colleagues [15] demonstrated the clinical efficacy and cost effectiveness of a universal scan in terms of detection of breech presentation in a cohort of 3,879 nulliparous women recruited to a research study, suggesting that undiagnosed term breech presentation could be "virtually eliminated" in these women, and leading to inferences that term breech presentation could be reduced. Our population-based cohort of over 27,000

women of all parities, assessing the "real world" impact of introducing such a policy, suggests that a universal scan policy may be less effective in clinical practice.

Why the universal scan is not followed by elimination of undiagnosed breeches is largely due to the presence of multiparous women who make up more than 50% of most pregnant populations. In nulliparous women, there were only 4 undiagnosed breeches. Although reversion after ECV is not more common among multiparous women [10], spontaneous version is [13], and the Wastlund study [15] only comprised nulliparous women. It is not as if these women are easily identifiable as, for instance, "unstable": Most undiagnosed breech babies were cephalic at the 36-week scan. This suggests it will be difficult to eliminate the undiagnosed term breech within existing resources or current scanning protocols.

The reduction in undiagnosed breech nevertheless means that there were only 2 vaginal breech births among 17 undiagnosed breeches after the introduction of the scan policy, approximating to 1/5,000 births. Vaginal breech birth may, however, be unavoidable where it is undiagnosed, and very late CS may be inadvisable [16] and cause harm [17]. Or vaginal birth may be preferred by the mother, as is the case in some European countries. The rarity of vaginal breech birth could further compromise skills for these occasions and limit maternal choice. Further, undiagnosed breech presentation will occur more often than this analysis suggests because of preterm birth, where breech presentation is much more common and labour progress can be rapid. It therefore remains essential that skills in vaginal breech birth are taught and maintained.

Why emergency CS reduced little following the introduction of a universal scan is difficult to explain. A reduction occurred among undiagnosed breech babies but not overall, and elective CS was indeed more frequent. It might have been because spontaneous labour occurred in the interval between booking the CS and the date of CS. This was not however because the elective CS had been booked late: There were no differences in gestation at birth, and there is no evidence that ECV promotes labour [16].

The introduction of a policy of a routine scan at 36 weeks for all singleton women is also not followed by a reduction in the incidence of breech presentation at term. It would be anticipated from the data in Wastlund and colleagues [15] that increased antenatal diagnosis would allow more breech babies to undergo ECV, so reducing the proportion of term babies who were breech. However, these authors could not realistically assess this crucial potential benefit because ECV was only attempted in less than half of their breeches and was successful in only 14.3% of attempts. In our cohort, ECV was more systematically used (>80%) and was much more successful, at 47.2% overall, a figure that is in line with our previously published data [10]. Our failure to reduce the incidence of breech presentation therefore requires explanation.

The UK RCOG Green Top Guideline [16] states "the greatest impediment to the use of ECV is the non-identification of breech presentation", yet our increased antenatal diagnosis did not translate into a reduced incidence at birth. Firstly, although ECV was contraindicated slightly less often, there were small but nonsignificant increases in both refusal and ECV failure rates. A reduction in success rate is plausible: Factors associated with a lower success rate of ECV, namely, a nonpalpable fetal head and a high BMI [18] are also likely to lower the sensitivity of clinical examination, and therefore reduce the impact of routine ultrasound scanning. Equally, where breech is proven by ultrasound before the ECV appointment, more time may be available to seek opinion and advice that might lead to a refusal to undergo the procedure. Nevertheless, the overall number of women who had undergone a successful ECV was slightly increased: As a percentage of the entire term cohorts, this is 1.0% and 1.2% of all pregnancies before and after the introduction of the scan, respectively (RR 1.20; 95% CI 0.96, 1.51). There are many possible explanations. Only 1 reversion occurred. It is unlikely that more babies were

delivered by the feet or breech where an elective CS was being performed anyway. The proportion of nulliparous women, in whom breech is more common, did not increase. The most likely explanation is due to the small but important number of breech babies that spontaneously turn to cephalic after 36 to 37 weeks. While the gestation age at the time of the ECV was not earlier after the universal scan policy, it is likely that at least some of the babies undergoing successful ECV would have spontaneously verted anyway. This does not mean that ECV is not effective: Meta-analysis of randomised controlled trials has shown it to reduce the number of breech presentations at birth [8]. It is merely that a proportion of breech babies turn anyway [13].

The influence of ECV on our interpretation is important. While the increased opportunity for ECV did not translate into fewer term breech presentations, the incidence of breech presentation was already very low, at 2.6%. This is likely to be because of relatively high existing detection rates and the comprehensive and successful existing ECV service. Our findings therefore only apply to units with good ECV services. The introduction of a universal scan where breech detection rates are low or where the initiative also involves an ECV service could reasonably reduce the incidence of term breech presentation.

We acknowledge other limitations in this study. Even with more than 27,000 births, we lack numbers to detect a small but genuine effect on morbidity or mortality. We analyse a surrogate for babies that were breech in the third trimester, rather than those that were breech at birth in order to be able to assess any impact of ECV on safety: Where a third trimester scan was not done, this will be inaccurate. Equally, our findings regarding mode of actual birth are influenced by offering vaginal breech birth, a policy which varies according to countries and individual units. We do not have more detailed maternal outcomes which would have been useful given the change in timing of CS birth. Finally, we acknowledge small but significant changes over the 4 years of the study: fewer births overall, increasing maternal age, and BMI. As this was not a randomised controlled trial, we cannot rule out that these, or other factors that we have not analysed, affected our results. These changes are not unique to OUH and reflect a changing demography [19]. However, there are strengths to the study design: In assessing the real world impact of a clinical implementation across an entire maternity unit, we are able to take account of factors such as these and of human and organisational factors that may not be apparent among research recruits.

## Conclusions

The introduction of a universal late third trimester ultrasound does not necessarily lead to a reduction in the incidence of term breech presentation. Although much less common, undiagnosed breeches at term remain. Clinical skill, both for the detection and the management of vaginal breech birth, continue to remain important.

## Supporting information

**S1 Analysis plan. Prespecified outcomes.**
(DOCX)

**S1 STROBE Checklist. STROBE checklist for cohort studies.**
(DOC)

**S1 Data. Study data set.**
(XLSX)

## Acknowledgments

We would like to acknowledge Dr Angelo Cavallaro, Anita Hedditch, the Breech Team, and all sonographers who contributed to this project.

## Author Contributions

**Conceptualization:** Ibtisam Salim, Lawrence Impey.

**Data curation:** Ibtisam Salim, Sam Mathewlynn, Lior Drukker.

**Formal analysis:** Ibtisam Salim, Eleonora Staines-Urias.

**Funding acquisition:** Lawrence Impey.

**Investigation:** Ibtisam Salim, Lawrence Impey.

**Methodology:** Ibtisam Salim, Sam Mathewlynn, Manu Vatish, Lawrence Impey.

**Project administration:** Sam Mathewlynn, Lawrence Impey.

**Supervision:** Manu Vatish, Lawrence Impey.

**Validation:** Ibtisam Salim, Sam Mathewlynn, Lior Drukker, Manu Vatish, Lawrence Impey.

**Visualization:** Ibtisam Salim, Lawrence Impey.

**Writing – original draft:** Ibtisam Salim, Eleonora Staines-Urias, Lawrence Impey.

**Writing – review & editing:** Ibtisam Salim, Eleonora Staines-Urias, Sam Mathewlynn, Lior Drukker, Manu Vatish, Lawrence Impey.

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
