## [Editor Report · Decision Letter 0]

24 Jan 2020

Dear Dr Salim, 

Thank you for submitting your manuscript entitled "The impact of a routine late third trimester growth scan on the incidence, diagnosis and management of breech presentation" for consideration by PLOS Medicine.

Your manuscript has now been evaluated by the PLOS Medicine editorial staff and I am writing to let you know that we would like to send your submission out for external peer review.

Please re-submit your manuscript within two working days, i.e. by 28 Jan 2020, 11:59PM.

Kind regards,

Louise Gaynor-Brook, MBBS PhD

Associate Editor, PLOS Medicine

---

## [Decision Letter · Decision Letter 1]

21 Aug 2020

Dear Dr. Salim,

Thank you very much for submitting your manuscript "The impact of a routine late third trimester growth scan on the incidence, diagnosis and management of breech presentation" (PMEDICINE-D-20-00199R1) for consideration at PLOS Medicine. We do apologize for the long delay in sending you a decision. 

Your paper was evaluated by an academic editor with relevant expertise, and sent to independent reviewers, including a statistical reviewer. The available reviews are appended at the bottom of this email - one further reviewer has seen your paper, and we will forward his or her report on to you via email if/when it becomes available - and any accompanying reviewer attachments can be seen via the link below:

[LINK]

In light of these reviews, we will not be able to accept the manuscript for publication in the journal in its current form, but we would like to invite you to submit a revised version that addresses the reviewers' and editors' comments fully. You will appreciate that we cannot make a decision about publication until we have seen the revised manuscript and your response, and we expect to seek re-review by one or more of the reviewers, to include a new reviewer if needed. 

We hope to receive your revised manuscript by Sep 11 2020 11:59PM. Please email us (plosmedicine@plos.org) if you have any questions or concerns.

Please let me know if you have any questions. Otherwise, we look forward to receiving your revised manuscript in due course. 

Sincerely,

Richard Turner PhD, for Louise Gaynor-Brook, MBBS PhD

Associate Editor, PLOS Medicine

rturner@plos.org

You mention that all relevant data are included - are you able include patient-level data in supplementary files, for example?

Please quote the setting in the title (e.g., "in Oxford, UK") and add a study descriptor (e.g., "a cohort study") following a colon. 

Please restructure the abstract to generate a combined "Methods and findings" section according to journal style. The final sentence of the new combined subsection should quote 2-3 of the study's main limitations. 

In your abstract, please quote aggregate demographic details for study participants. 

After the abstract, please add a new and accessible "Author summary" section in non-identical prose. You may find it helpful to consult one or two recent research papers published in PLOS Medicine to get a sense of the preferred style. 

Early in the methods section of your main text, please state whether or not the study had a protocol or prespecified analysis plan (and if so attach the relevant document(s) as supplementary file(s)). Please highlight analyses that were not prespecified. 

We notice an ethics approval date in 2017 quoted in the methods section - please explain how this relates to the date the study started. Also, we suggest noting the situation regarding participant informed consent - perhaps not required in routine care. 

In table 1, we suggest "White" rather than "Caucasian" ethnicity. 

Please adapt the first paragraph of your "Discussion" section to provide a summary of the study findings - i.e., employing wording such as "we found that ... was" rather than the more general "shows this is not the case", for example, with the latter moved to a subsequent paragraph if you wish. 

In the "Conclusion" section at the end of the main text, please adapt the text to "In this study, we found that ... did not lead ..." to avoid over-generalizing conclusions from a single-centre study. 

Please quote p values alongside 95% CI throughout the paper, where available. 

Throughout the text, please format reference call-outs as follows: "... preterm births [1].".

Please remove all iterations of "[Internet]" from the reference list. 

Please add a completed checklist for the most appropriate reporting guideline, which we suspect will be STROBE, as a supplementary document (referred to in the methods section, e.g., "See S1_STROBE_Checklist"). In the checklist please refer to individual items by section (e.g., "Methods") and paragraph number rather than by line or page numbers, as the latter generally change in the event of publication. 

Comments from the reviewers:

*** Reviewer #1: 

This article is about the impact that a change in procedure to routinely have late trimester scans has on various outcomes including breech presentation. 

As a statistical reviewer, there is very little statistical considerations being reported in the article for me to be able to comment on, and only basic summary statistics and unadjusted two-sample tests are used to compare between the populations before and after the change in procedure. Similarly, given the methods of data collection there is very little on study design or methods that is relevant to comment on. Perhaps the only comment I would have for the statistics, is that the method of calculation for the Risk Ratio confidence interval should be stated.

The above comments are not implying that the article is not of value or that further statistics/analysis would be needed to answer the key question of interest, just that there are only very basic numbers reported and so the value of the paper will almost entirely be based on clinical judgement and relevance for which I couldn't reasonably contribute towards. 

*** Reviewer #2: 

General and specific comments included below. This paper is highly topical to an obstetric/maternity care audience, as the role of routine third trimester ultrasound continues to be considered in many developed settings worldwide.

There are already a few major papers looking at the identification of breech babies via routine T3 US and the consequent impact on care/outcomes - this paper attempts to investigate whether routine T3 ultrasound results in changes in antenatal detection, mode of birth and incidence of breech presentation in a 'real world' setting (primips and multips).

I think the premise is good and an important question but I think the findings that form the focus of this paper are not those that are most clinically relevant or significant. The paper concludes that routine T3 US did not reduce the presentation of breech at term but for a clinical audience, it is much more interesting and salient that routine US did significantly reduce the incidence/risk of undiagnosed breech before birth and labour - this is where the majority of breech-related morbidity lies so it would seem to be of great significance but is not how this paper is framed. I would suggest reframing the discussion of this paper if it is intended to be published for a clinical audience.

Abstract 

In the conclusion, an erroneous 'of'

Intro

States CS is 'more costly' - than what? Vaginal birth? Because elective CS for breech is arguably much less costly than breech vaginal birth. This needs to be clarified.

The paper states that the greatest negative of breech presentation at term is the contribution to CS rate but the key end-point/question of the study is incidence of breech presentation at term. Breech at term doesn't really matter in the clinical setting unless it changes a clinical outcome, such as emergency CS rate for undiagnosed breech specifically or unplanned vaginal breech births.

Methods

Only one line given about determining RR - how was this done statistically? This would seem a good study in which to use causal inference methodology and attempt to replicate the assumptions of an RCT, especially given the clear exposure of universal US and known differences in the demographic characteristics of pre and post US cohorts.

Results

Summary of key findings:

No change in term breech presentation (S)

More breech presentation at clinic (S)

More planned breech VD (NS)

Fewer actual breech VD (S)

More elective CS overall (S)

Fewer emergency CS overall (NS)

Significant reduction in undiagnosed breech before birth and before labour - these are narrow CI and surely clinically significant findings!

Hard to work out how (Table 1) gestational age at delivery could be significantly different between the pre and post-OxGRIP groups given numbers provided?

Table 1 reports breech at delivery as 2.6 vs 2.7 and the text states breech at term is the same - these are potentially different things, needs to be clarified if being used interchangeably.

Planned vaginal breech increased but actual vaginal breech reduced overall (not in those in whom breech was diagnosed) - Was this because X% of planned breech births reverted to cephalic before birth or because they were delivered by CS??

Table 3 indicates that undiagnosed breech before birth was significantly reduced from 25.5% to 8.3% and that amongst the undiagnosed breech babes at term, breech undiagnosed before labour was significantly reduced - this seems to be an important clinical end-point.

Error - 'the proportion of breech babies giving birth' does not make sense - 'being born vaginally' perhaps?

Also, 'a surrogate for babies that were breech in the third trimester babies' - an extra 'babies'?

It would be informative to know whether those two breech babies that died pre-OxGRIP were born via vaginal birth or were undiagnosed breech even?

The study is likely under-powered to detect any meaningful reductions in adverse perinatal outcomes but still demonstrates a trend towards reduction across all outcomes - seems important in the context of the study.

Discussion

The authors seem to emphasise the findings of the study that are not of greatest clinical significance. The steady rate of term breech presentation pre and post routine T3 US is not that interesting, but the significant reduction in actual vaginal breech births and in undiagnosed breech presentation at birth or pre-labour is of great clinical significance and will get the attention of a clinical readership - would seem to justify the intervention. Further, although not powered to determine significance, the trend towards better neonatal outcomes, though rare, is also meaningful for clinicians. This is the main reason to care about breech presentation at term and therefore the role of routine T3 US, not so much the change in breech presentation or caesarean section rates.

Error: First paragraph says 'at last' instead of 'at least'

Error: 'Whilst the gestation at the time of the ECV gestation at ECV' - sentence does not make sense

***

[LINK]

---

## [Decision Letter · Decision Letter 2]

26 Oct 2020

Dear Dr. Salim,

Thank you very much for submitting your revised manuscript "The impact of a routine late third trimester growth scan on the incidence, diagnosis and management of breech presentation in Oxford, UK: A cohort study" (PMEDICINE-D-20-00199R2) for consideration at PLOS Medicine. 

Your paper was discussed with our academic editor and re-seen by two reviewers, and by a new statistical reviewer. The reviews are appended at the bottom of this email and any accompanying reviewer attachments can be seen via the link below:

[LINK]

In light of these reviews, we will not be able to accept the manuscript for publication in the journal in its current form, but we would like to invite you to submit a further revised version that addresses the reviewers' and editors' comments fully. You will recognize that we cannot make a decision about publication until we have seen the revised manuscript and your response, and we plan to seek re-review by one or more of the reviewers. 

We hope to receive your revised manuscript by Nov 16 2020 11:59PM. Please email us (plosmedicine@plos.org) if you have any questions or concerns.

Please let me know if you have any questions. Otherwise, we look forward to receiving your revised manuscript shortly. 

Sincerely,

Richard Turner PhD, for Louise Gaynor-Brook, MBBS PhD

Associate Editor, PLOS Medicine

rturner@plos.org

Bearing in mind the observational design of your study, please adapt wording throughout your paper which ascribes causality to the policy change. For example, at line 44 we ask you to amend the wording to "In this study ... was associated with a reduction in the incidence of undiagnosed breech ...". 

Similarly at line 270, we suggest amending the text to "It is difficult to explain why the policy was associated with a non-significant reduction in emergency CS. While there was an apparent fall in the number of undiagnosed breech babies but not overall ...", or similar. 

In the title, would "before and after study" be a more appropriate study descriptor than "cohort study"?

Please harmonize Oxford/Oxfordshire in title and paper. In the abstract, we suggest adapting the text to note that the study was done at a single centre. 

Please make that "principal" at line 20. 

Please refer to the attached study plan early in the methods section of your main text, and highlight any non-prespecified analyses. 

Please use the notation "p<0.001" throughout. 

We suggest removing the attached RCOG abstract. 

In the attached study plan, we suggest deleting the text about the "helper second author". 

Are you able to include participant-level data in supplementary files, or deposit these at a publicly-accessible repository?

Comments from the reviewers:

*** Reviewer #2: 

The authors have made considerable effort to incorporate the reviewers feedback and comments and the resulting manuscript is clearer and more compelling as a result. The paper present a very large cohort study with interesting findings and I feel it should be published. This iteration of the paper presents a much more clinically-relevant perspective and is much easier to comprehend - the first version was confusing in parts between what was and was not impacted by the US screening intervention. The discussion is now much more balanced and fair, and presents reasonable recommendations for other units based on the findings and limitations of this study.

*** Reviewer #3: 

Having reviewed the authors' response and the submitted new version of The impact of a routine late third trimester growth scan on the incidence, diagnosis and management of breech presentation in Oxford, UK: A cohort study" (PMEDICINE-D-20-00199R2) by Dr Ibtisam Salim and colleagues for PLOS Medicine I think it is suitable for publication in Plos Med.

Authors responded well to the suggestions of my comments and the second reviewer. The manuscript is clear, only the 2 pre-specified cohorts (pre vs post scan) are used and outcome is well-defined. The new stated conclusion that a routine ultrasound considerably reduced the incidence of undiagnosed term breech presentation at birth is clear, even though this did not result in no undiagnosed breech presentation, a reduction in the incidence of breech presentation and better neonatal outcome. I agree fully with their comment that it is extremely important to show that the introduction of a third trimester scan does not reduce the incidence of breech presentation at delivery, particularly in a unit with widespread and successful use of ECV and therefore obstetricians with clinical skills for vaginal breech delivery should remain. 

*** Reviewer #4: 

I confine my remarks to statistical areas of this paper. I have a couple of issues to resolve before I recommend publication

First, there is no reason to do the tests on lines 137-138. For "table 1" analysis the key is effect size, not significance. Even very small differences were significant here, but not meaningful. Also, eliminate the p value column in table 1.

Second, and more importantly, while it isn't exactly *wrong* to do t tests as the authors did, it would be better to use logistic regression and account for covariates. This will increase the staistical power of the tests, and also allow some other conclusions to be drawn.

Peter Flom

***

[LINK]

---

## [Decision Letter · Decision Letter 3]

25 Nov 2020

Dear Dr. Salim,

Thank you very much for re-submitting your manuscript "The impact of a routine late third trimester growth scan on the incidence, diagnosis and management of breech presentation in Oxford, UK: A cohort study" (PMEDICINE-D-20-00199R3) for consideration at PLOS Medicine.

I have discussed the paper with our academic editor and it was also seen again by one reviewer. I am pleased to tell you that, once the remaining editorial and production issues are dealt with, we expect to be able to accept the paper for publication in the journal.

[LINK]

Please let me know if you have any questions. Otherwise, we look forward to receiving the revised manuscript shortly. 

Kind regards,

Richard Turner PhD, for Louise Gaynor-Brook, MBBS PhD

Associate Editor, PLOS Medicine

rturner@plos.org

Requests from Editors:

We suggest making that "Oxfordshire" in the title. 

At line 26, please adapt the text to "We carried out a population-based ..." or similar.

Please restructure the end of the "Methods and findings" subsection of you abstract. The final sentence should begin "Study limitations include ..." or similar, and should list 2-3 limitations. We suggest that one limitation should be the possible influence of secular changes over time on your findings. 

The final sentence of the "Conclusions" subsection of your abstract seems rather disjointed, and we suggest running on the previous sentence, e.g., "... service, indicating that clinical skills ...". Alternatively the sentence could be removed. 

At line 63, we suggest quoting the actual number of births.

Please remove the quotation marks at lines 63 and 116/7.

At line 72, please amend the text to "... is unlikely to prevent the occasional need for skills ...". 

At line 77, we suggest adapting the text to "... reduced expertise in management of those ...".

Please begin the sentence at line 239 with "We observed that ..." or similar. 

At line 246 and any other instances in the ms, we ask you to amend the word "trend". Please substitute a phrase such as "apparent non-significant benefit". 

At line 323, we suggest substituting "rule out" for "disprove".

At line 325, please adapt the text to "However, there are also strengths to the study design: ...".

Please remove the information on funding from the acknowledgements section at the end of the main text. This information will appear in the article metadata via entries in the submission form. 

Throughout the text, the labels for tables seem incorrect; e.g., at line 192, "S1_Table" should be "Table 1".

Please ensure that the attachments (e.g., STROBE checklist) are labelled as they are referred to in the text. 

Comments from Reviewers:

*** Reviewer #3: 

The authors have responded to my suggestions.

I suggested removing the p values from Table 1. They said they would like to keep them, or better understand why I asked to remove them. Here is that explanation:

It isn't wrong to put the p values; that is, it doesn't violate any rules or assumptions or whatnot. But it is misleading. It perpetuates myths about p values. Just as, say, you wouldn't include meaningless medical information (like, say, patient's hair color) so you shouldn't include p values. P values tell you whether to reject a null hypothesis. But, here, there is no null hypothesis. The key issue is whether the two groups are different in any meaningful way. That is answered by the effect sizes. The fact that, e.g. the change in mean maternal age is significant isn't really relevant. It went up by 0.3 years. Unless there is some reason that that difference is of medical significance, why put in the p values?

But I won't insist on the removal, I just think it is a good idea.

I also suggested using logistic regression instead of t-tests. The authors responded that they felt they did not need to do so because the differences on the covariates were small; they wrote

<<<

The covariates for which adjustment might be meaningful would be those that might reasonably have influenced the diagnosis or management of breech resentation, whilst not being on the causal pathway whereby the intervention (a universal scan) is likely to have led to the changes observed.

>>>

This isn't necessarily so. Even if a variable is equal in the two groups, it can still change the parameter estiamates for the main independent variable. It can also decrease the standard error. 

Again, I won't insist on the change because, as I noted originally, it isn't exactly wrong to do t-tests. But, again, I think logistic reg. would be better.

So, I marked "proceed without recommendation" and leave it to the authors and editors how to proceed.

Peter Flom

***

[LINK]

---

## [Editor Report · Decision Letter 4]

16 Dec 2020

Dear Dr. Salim,

I am writing concerning your manuscript submitted to PLOS Medicine, entitled “The impact of a routine late third trimester growth scan on the incidence, diagnosis and management of breech presentation in Oxfordshire, UK: A cohort study.”

We have now completed our final technical checks and have approved your submission for publication. You will shortly receive a letter of formal acceptance from the editor.

Kind regards,

PLOS Medicine